# GRASPEL: GRAPH SPECTRAL LEARNING AT SCALE

## ABSTRACT

Learning meaningful graphs from data plays important roles in many data mining and machine learning tasks, such as data representation and analysis, dimension reduction, data clustering, and visualization, etc. In this work, for the first time we present a highly-scalable spectral approach (GRASPEL) for learning large graphs from data. By limiting the precision matrix to be a graph Laplacian, our approach aims to estimate ultra-sparse (tree-like) weighted undirected graphs and shows a clear connection with the prior graphical Lasso method. By interleaving the latest high-performance nearly-linear time spectral methods for graph sparsification, coarsening and embedding, ultra-sparse yet spectrally-robust graphs can be learned by identifying and including the most spectrally-critical edges into the graph. Compared with prior state-of-the-art graph learning approaches, GRASPEL is more scalable and allows substantially improving computing efficiency and solution quality of a variety of data mining and machine learning applications, such as spectral clustering (SC), and t-Distributed Stochastic Neighbor Embedding (t-SNE). For example, when comparing with graphs constructed using existing methods, GRASPEL achieved the best spectral clustering efficiency and accuracy.

## 1 INTRODUCTION

Graph construction is playing increasingly important roles in many machine learning and data mining applications. For example, a key step of many existing machine learning methods requires converting potentially high-dimensional data sets into graph representations: it is a common practice to represent each (high-dimensional) data point as a node, and assign each edge a weight to encode the similarity between the two nodes (data points). The constructed graphs can be efficiently leveraged to represent the underlying structure of a data set or the relationship between data points (Jebara et al., 2009; Maier et al., 2009; Liu et al., 2018). However, how to learn meaningful graphs from large data set at scale still remains a challenging problem.

In the past decades, considerable effort has been devoted to the development of graph construction methods. For example, constructing $k$-nearest-neighbor (kNN) graphs requires each node to be connected with its top-$k$ nearest neighbors, while in construction of the $\epsilon$-neighborhood graphs all the neighbors within the range of distance $\epsilon$ will be connected; to improve the capability of kNN graph in handling multi-scale data, (Zelnik-Manor & Perona, 2005) introduced a self-tuning technique to adjust the local scaling parameter for similarity measurement; to find meaningful similarity measures between nodes, (Bach & Jordan, 2006) propose to learn the similarities from feature vectors in a supervised setting; (Zhu et al., 2014) adopted an information-theoretic definition of data similarity to capture subtle similarity information; (Jebara & Shchogolev, 2006) proposed to remove spurious edges from kNN graph via b-matching; (Pavan & Pelillo, 2007) introduced a method for removing noisy edges by selecting the maximum cliques; (Premachandran & Kakarala, 2013) proposed to leverage collected consensus information form various neighborhoods to improve the robustness of the kNN graph; (Nie et al., 2014) proposed to learn the adjacency graph by adaptively assigning neighbors. However, the aforementioned nearest-neighbor (NN) based graph construction methods can only capture local manifold information and may not be able to truthfully reveal the global structure of a given data set (Nie et al., 2016; Liu et al., 2018; Guo, 2015), which can result in over complicated (with too many edges) or sometimes misleading graph representations. For example, choosing different numbers of nearest neighbors for constructing kNN graphs may lead to drastically different classification performance in spectral clustering tasks (Chen et al., 2018).

Several recent graph learning methods leverage emerging graph signal processing (GSP) techniques for estimating sparse graph Laplacians, which show very promising results (Dong et al., 2016; Egilmez et al., 2017; Dong et al., 2019; Kalofolias & Perraudin, 2019). For example, (Egilmez et al., 2017) addresses the graph learning problem by restricting the precision matrix to be a graph Laplacian and maximizing a posterior estimation of Gaussian Markov Random Field (GMRF), while an $L1$-regularization term is used to promote graph sparsity; (Rabbat, 2017) provides an error analysis for

inferring sparse graphs from smooth signals; (Kalofolias & Perraudin, 2019) leverages approximate nearest-neighbor (ANN) graphs to reduce the number of variables for optimization. However, even the state-of-the-art Laplacian estimation methods for graph learning do not scale well for large data set due to their extremely high algorithm complexity. For example, solving the optimization problem for Laplacian estimation in (Dong et al., 2016; Kalofolias, 2016; Egilmez et al., 2017; Dong et al., 2019) requires $O(N^2)$ time complexity per iteration for $N$ data entities and nontrivial parameters tuning for controlling graph sparsity which limits their applications to only very small data sets (e. g. with up to a few thousands of data points). The latest graph learning approach (Kalofolias & Perraudin, 2019) takes advantages of ANN graphs but can still run rather slowly for large data sets.

This work for the first time introduces a spectral method (GRASPEL) for learning ultra-sparse graphs from data by leveraging the latest results in spectral graph theory (Feng, 2016; 2018; Zhao et al., 2018). There is a clear connection between our approach and the GSP-based Laplacian estimation methods (Dong et al., 2016; Kalofolias, 2016; Egilmez et al., 2017; Kalofolias & Perraudin, 2019; Dong et al., 2019), as well as the classical graphical Lasso framework (Friedman et al., 2008). Specifically, by treating $p$-dimensional data points as $p$ graph signals, GRASPEL learns a graph Laplacian by maximizing its first few eigenvalues as well as the smoothness of graph signals across edges, subject to a graph sparsity constraint. By iteratively interleaving recent nearly-linear time spectral graph sparsification, coarsening and embedding methods (Feng, 2016; 2018; Zhao et al., 2018), GRASPEL enjoys a nearly-linear runtime and space complexity.

GRASPEL is similar to the original graphical Lasso (Friedman et al., 2008) with the precision matrix replaced by a graph Laplacian. GRASPEL iteratively identifies and includes the most spectrally-critical edges into the latest graph, so that the first few Laplacian eigenvalues and eigenvectors can be most significantly perturbed by adding the minimum amount of edges. The iterative graph learning procedure will be terminated when the graph spectra become sufficiently stable (or graph signals become sufficiently smooth across the graph and lead to rather small Laplacian quadratic forms). Comparing with state-of-the-art methods, GRASPEL allows more scalable estimation of attractive Gaussian Markov Random Fields (GMRFs) for even very large data set. We show through extensive experiments that GRASPEL can learn high-quality ultra-sparse (tree-like) graphs that can be immediately leveraged to significantly improve the efficiency and accuracy of spectral clustering (SC) tasks; the proposed approach also leads to the development of a multilevel t-Distributed Stochastic Neighbor Embedding (t-SNE) algorithm that shows significantly improved runtime over existing methods (Maaten & Hinton, 2008; Van Der Maaten, 2014).

## 2 BACKGROUND OF GRAPH LEARNING VIA LAPLACIAN ESTIMATION

Given $M$ observations on $N$ data entities stored in a data matrix $X \in \mathbb{R}^{N \times M}$, each column of $X$ can be considered as a signal on a graph. The recent graph learning method (Dong et al., 2016) aims to estimate a graph Laplacian from $X$ while achieving the following desired characteristics:

**1) Smoothness of signals on the graph.** The graph signals corresponding to the real-world data should be sufficiently smooth on the learned graph structure: the signal values will only change gradually across connected neighboring nodes. The smoothness of a signal $x$ over a undirected graph $G = (V, E, w)$ can be measured with Laplacian quadratic form $x^T L x = \sum_{(p,q) \in E} w_{p,q} (x(p) - x(q))^2$, where $L = D - W$ denotes the Laplacian matrix of graph $G$ with $D$ and $W$ denoting the degree and the weighted adjacency matrices of $G$, and $w_{p,q} = W(p, q)$ denotes the weight for edge $(p, q)$. The smaller value of quadratic form indicates the smoother signals across the graph. It is also possible to quantify the smoothness $(Q)$ of a set of signals $X$ over graph $G$ using the following matrix trace (Kalofolias, 2016): $Q(X, L) = Tr(X^T L X)$, where $Tr$ denotes the matrix trace.

**2) Sparsity of the estimated graph (Laplacian).** Graph sparsity is another critical consideration in graph learning. One of the most important motivations of learning a graph is to use it for downstream data mining or machine learning tasks. Therefore, desired graph learning algorithms should allow better capturing and understanding the global structure (manifold) of the data set, while producing sufficiently sparse graphs that can be easily stored and efficiently manipulated in the downstream algorithms, such as graph clustering, partitioning, dimension reduction, data visualization, etc. To this end, the graphical Lasso algorithm (Friedman et al., 2008) has been proposed to learn the structure in an undirected Gaussian graphical model using $L1$ regularization to control the sparsity of the precision matrix. Given a sample covariance matrix $S$ and a regularization parameter $\beta$, graphical Lasso targets the following objective function:

$$\max_{\Theta} \log \det(\Theta) - Tr(\Theta S) - \beta \|\Theta\|_1, \tag{1}$$

over all non-negative definite precision matrices $\Theta$. The first two terms together can be interpreted as the log-likelihood under a Gaussian Markov Random Field (GMRF). $\beta\|\Theta\|_1$ is the sparsity promoting regularization term. This model tries to learn the graph structure by maximizing the penalized log-likelihood. However, the log-determinant problems are very computationally expensive. The emerging GSP-based methods infer the graph by adopting the criterion of signal smoothness (Kalofolias, 2016; Dong et al., 2016; Egilmez et al., 2017; Kalofolias & Perraudin, 2019). However, their extremely high complexities do not allow for learning large-scale graphs involving millions or even hundred thousands of nodes. Furthermore, these methods usually require nontrivial parameters tuning for controlling graph sparsity.

## 3  GRASPEL: A SPECTRAL APPROACH TO GRAPH LEARNING FROM DATA

At high level, GRASPEL gains insight from recent GSP-based Laplacian estimation methods (Dong et al., 2019), aiming to solve the following optimization problem that is similar to the graphical Lasso problem (Friedman et al., 2008):

$$\textbf{maximize:}_{L \in \mathcal{L}} \ \log\det(L) - \alpha Tr(X^T L X) - \beta\|L\|_1, \tag{2}$$

where $\mathcal{L}$ denotes the set of valid Laplacian matrices. It can be shown that the three terms in (2) are corresponding to $\log\det(\Theta)$, $Tr(\Theta S)$ and $\beta\|\Theta\|_1$ in (1), respectively. When the precision matrix $\Theta$ is restricted to be a graph Laplacian, and each column vector in the original data matrix $X$ is treated as a graph signal vector, there is a close connection between our formulation and the graphical Lasso problem. Since graph Laplacians are symmetric and positive definite (PSD) matrices (or M matrices) with non-positive off-diagonal entries, this formulation will lead to the estimation of attractive GMRFs (Dong et al., 2019).

### 3.1  OVERVIEW OF GRASPEL

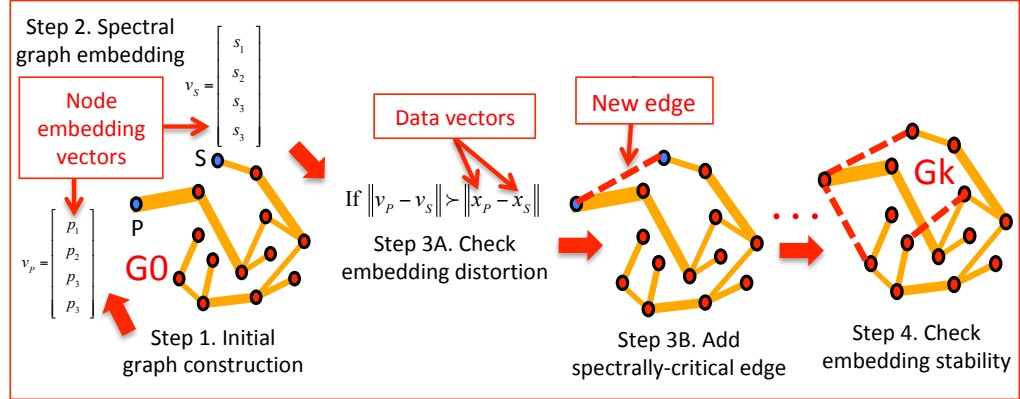

Figure 1: The overview of the proposed GRASPEL framework.

To achieve good efficiency in graph learning that may involve millions of nodes, instead of solving (2) directly, GRASPEL leverages a spectral approach for solving (2) implicitly. Define *spectrally-critical edges* to be the ones that can most effectively perturb the graph spectral properties, such as the first few Laplacian eigenvalues and eigenvectors. GRASPEL aims to iteratively identify and add the most spectrally-critical edges into the latest graph until no such edges can be found, which consists of the following key steps as illustrated in Figure 1:

**Step (1): Initial graph construction.** Similar to (Kalofolias & Perraudin, 2019), we start with constructing an ANN graph that can be achieved in $O(N \log N)$ time, where each edge weight encodes the similarity (e.g. Gaussian kernel or cosine similarity) of two data entities; next, the ANN graph is converted into an ultra-sparse nearest-neighbor (uNN) graph with $O(N \log N)$ edges by leveraging a nearly-linear-time spectral sparsification algorithm (Feng, 2018).

**Step (2): Spectral graph embedding.** We apply a nearly-linear-time spectral embedding procedure (Zhao et al., 2018) to the current graph so that each node will be associated with a low-dimensional embedding vector (e.g. $v_s$ for node $S$ in Figure 1), where the embedding dimension (number of eigenvectors) can be determined based on the largest gaps of the first few (e.g. 100) Laplacian eigenvalues (Peng et al., 2015).

**Step (3): Spectrally-critical edge identification.** We quickly check the *embedding distortion* for each candidate edge (node pair) defined as $\eta = \frac{z^{emb}}{z^{data}}$, where $z^{emb}$ ($z^{data}$) denotes the distance in the embedding (data) vector space. The edges with the largest $\eta$ are considered as the most spectrally-critical edges and will be added into the latest graph.

**Step (4): Spectral stability checking.** After repeating the Steps (2)-(3) multiple times for adding new edges, GRASPEL will return the final graph once the overall embedding distortion becomes sufficiently small or stable: if the first few Laplacian eigenvalues do not change much over iterations of adding extra edges, the graph spectra is considered stable (robust) since adding more edges does not significantly perturb the key (smallest) eigenvalues.

**Connection between GRASPEL and the formulation (2)**. The original optimization objective function includes the following three components: **(a)** $\log \det(L)$ corresponding to the sum of the logarithmic Laplacian eigenvalues, **(b)** $-\alpha Tr(X^T L X)$ corresponding to the smoothness of signals across the graph [1], and **(c)** $-\beta * |L|_1$ corresponding to graph sparsity. Including spectrally-critical edges into the graph will significantly impact the first few Laplacian eigenvalues and eigenvectors key to graph spectral properties, thereby dramatically improving embedding distortion and the overall smoothness of signals across the graph. It can be shown that including any additional edge into the graph will monotonically increase (a) while monotonically decreasing both (b) and (c). Therefore, when the spectra of the learned graph is not stable adding any spectrally-critical edges into the graph will dramatically increase (a) while slightly decreasing (b) and (c), since the improved graph signal smoothness will only result in a slight change (increase) to $Tr(X^T L X)$.

**Convergence analysis for the GRASPEL framework**. The objective function in (2) will be effectively maximized by including only a small amount of spectrally-critical edges until the first few eigenvalues become sufficiently stable; when adding extra edges can no longer perturb the first few eigenvalues, (b) and (c) will start to dominate the objective function value, indicating that the iterations should be terminated. The stopping condition can be controlled by properly setting an embedding distortion threshold for $\eta$ or parameters $\alpha$ and $\beta$.

**Complexity analysis for the GRASPEL framework**. To achieve scalable spectral graph embedding key to identification of spectrally-critical edges in Steps (2)-(3), we will leverage the latest high-performance spectral graph algorithms. Since all the kernel functions involved in GRASPEL, such as ANN graph construction (Muja & Lowe, 2009; 2014; Malkov & Yashunin, 2018), spectral graph sparsification (Feng, 2016; 2018), spectral coarsening and embedding (Zhao et al., 2018), are all nearly-linear $O(N \log N)$ time algorithms, the entire spectral graph learning approach GRASPEL also has a nearly-linear time complexity.

## 3.2 DETAILED STEPS IN GRASPEL

### 3.2.1 INITIAL GRAPH CONSTRUCTION

As aforementioned, (approximate) kNN graphs can be used to construct the initial graphs in Step (1), since they can be created very efficiently (Muja & Lowe, 2009), while being able to approximate the local data proximity (Roweis & Saul, 2000). However, traditional kNN graphs have the following drawbacks: **a)** the kNN graphs with large $k$ (the number of nearest neighbors) has the tendency of increasing the cut-ratio (Qian et al., 2012); **b)** the optimal $k$ value is usually problem dependent and can be very difficult to find. In this work, we will start creating an (approximate) kNN graph with a relatively small $k$ value (e.g. $k = 5$), and strive to significantly improve the graph quality by adding extra spectrally-critical edges through implicitly solving the proposed optimization problem in (2). In the last, a spectral sparsification algorithm (GRASS) [2] has been applied to further simplify the kNN graph into a uNN graph with only $O(N \log N)$ edges (Wang & Feng, 2017).

### 3.2.2 SPECTRAL GRAPH EMBEDDING

Spectral graph embedding directly leverages the first few nontrivial eigenvectors for mapping nodes onto low-dimensional space (Belkin & Niyogi, 2003). The eigenvalue decomposition of Laplacian matrix is usually the computational bottleneck in spectral graph embedding, especially for large graphs (Shi & Malik, 2000; Von Luxburg, 2007; Chen et al., 2011). To achieve good scalability, we exploit multilevel spectrally-reduced graphs that allow for much faster eigenvector (eigenvalue) computations without loss of accuracy (Zhao et al., 2018). Specifically, the multilevel method first

---

[1]When graphs signals in $X$ are sufficiently smooth, they will align well with the first few eigenvectors corresponding to the smallest few eigenvalues, leading to relatively small trace $Tr(X^T L X)$.

[2]GRASS can be downloaded at https://sites.google.com/mtu.edu/zhuofeng-graphspar

spectrally coarsens the fine-level graph into much smaller ones with preservation of key spectral properties, and then maps the eigenvectors obtained on the coarse graphs back to the original graph; multilevel eigenvector refinement (smoothing) and orthogonalization procedures can be applied to further improve the approximation accuracy (Zhao et al., 2018).

### 3.2.3 SPECTRALLY-CRITICAL EDGE IDENTIFICATION

Once Laplacian eigenvectors are available for the current graph, we can identify spectrally-critical edges by looking at each candidate edge's embedding distortion ($\eta$). To this end, we exploit the following first-order spectral perturbation analysis to quantitatively evaluate each candidate edge's impact on the first few eigenvalues. Denote the edge weight by $w_{p,q}$ and the Laplacian eigenvector corresponding to the eigenvalue $\lambda_i$ by $u_i$. We define $e_p \in \mathbb{R}^n$ to be a standard basis vector with all zero entries except for the $p$-th being 1, and $e_{p,q} = e_p - e_q$. The following theorem will allow us to identify the most spectrally-critical edges leveraging the first few Laplacian eigenvectors.

**Theorem 1** *The spectral criticality $c_{p,q}$ or embedding distortion $\eta_{p,q}$ of a candidate edge $(p,q)$ on the Laplacian eigenvalue $\lambda_i$ can be properly estimated by $c_{p,q} = w_{p,q} \left( u_i^T e_{p,q} \right)^2 \propto \eta_{p,q} = \frac{z_{p,q}^{emb}}{z_{p,q}^{data}}$.*

Proof: See the Appendix.

**Edge identification with Fiedler vectors.** The idea for identifying spectrally-critical edges is to sort nodes according to the Fiedler vector. Only the node pairs with large embedding distances will be examined as candidate edges. Therefore, we are able to limit the search within the candidate edge connections between the top and bottom few nodes in the $1D$ sorted node vector. Only the candidate edges with top spectral criticality or embedding distortion values will be added into the latest graph.

**Edge identification with multiple eigenvectors.** With $k$ eigenvectors for spectral embedding, we can first project the graph nodes onto a $k$-dimensional space and perform spectral clustering to group the nodes into $k$ clusters, where the embedding dimension $k$ can be determined based on the largest gaps of the first few (e.g. 100) Laplacian eigenvalues (Peng et al., 2015). Next, we only have to examine the candidate edges that connect nodes between two distant clusters in the embedding space, and sort them based on embedding distortions.

To further reduce the computational cost, we first perform spectral graph coarsening (Zhao et al., 2018) and subsequently search for high-distortion candidate edges on the coarsest graph. Once a small set of top spectrally-critical edges has been identified, we will find their corresponding candidate edges in the original graph. Since each coarse-level candidate edge may correspond to multiple candidate edges in the original graph, we will sort them based on their embedding distortions, and only add the ones with largest distortions into the latest graph. The algorithms for spectrally-critical edge identification using Fiedler vector and multiple eigenvectors have been described in Algorithm 1 and Algorithm 2 in the Appendix.

### 3.2.4 SPECTRAL STABILITY CHECKING

We employ an iterative procedure in GRASPEL to repeatedly add new edges into the graph, thereby improving the approximation quality. Specifically, at each iteration new spectrally-critical edges are identified and added to the current graph; we will terminate the iterations when the graph spectra become sufficiently stable. Alternatively, we can check if the embedding distortions of top candidate edges still keep improving; if not, the iterations can be terminated, indicating that no additional spectrally-critical edges can be found to significantly perturb the first few Laplacian eigenvectors and eigenvalues. In practice, we observe that the number of spectrally-critical edges decreases rapidly within very few (two to four) iterations.

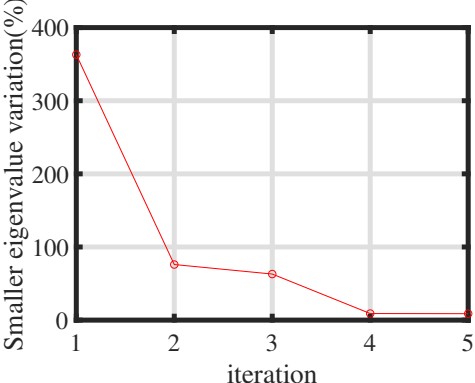

Figure 2: Variation ratio of bottom eigenvalues with increasing number of iterations.

For GRASPEL we adopt the following scheme for checking the spectral stability of each graph learning iteration: 1) in each iteration, we compute and record the several smallest eigenvalues of the latest graph Laplacian according to the largest gap between eigenvalues (Peng et al., 2015): for example, the first (smallest) $k$ nonzero eigenvalues that are critical for spectral clustering will be stored; 2) we check

whether a sufficiently stable spectra has been reached for graph learning by comparing them with the eigenvalues computed in the previous iteration: if the change is significant, more iterations may be needed. To this end, we record the first $k$ Laplacian eigenvalues computed in the previous (current) iteration into vector $v_p$ ($v_{p+1}$), and calculate the spectral variation ratio by:

$$ratio_{var} = \frac{\|v_p - v_{p+1}\|}{\|v_p\|}. \tag{3}$$

A greater spectral variation ratio indicates less stable eigenvalues within the latest graph, and thus justifies another iteration for adding more spectrally-critical edges into the graph. The spectral stability checking results for the USPS data set (see Appendix for details) have been shown in Figure 2; as observed, only four iterations will suffice for achieving a rather stable graph spectra.

## 4 Experiments

In this section, extensive experiments have been conducted to evaluate the performance of GRASPEL for a variety of public domain data sets. Note that all the graphs learned by GRASPEL have ultra-sparse tree-like structures with graph densities (defined as $|E|/|V|$) between 1.1 to 1.3; our approach allows learning much sparser graphs when comparing with the latest approach that always produces graphs with densities much greater than 3.0 (Kalofolias & Perraudin, 2019).

### 4.1 Graph learning for spectral clustering

As shown in Algorithm 3 in Appendix, the the classical spectral clustering (SC) algorithm first constructs a graph where each edge weight encodes similarities between different data points (entities); then SC calculates the eigenvectors of the graph Laplacian matrix and embeds data points into low-dimensional space (Belkin & Niyogi, 2003); in the last, k-means algorithms are used to partition the data points into multiple clusters. The performance of SC strongly depends on the quality of the underlying graph (Guo, 2015). In this section, we apply GRASPEL for graph construction, and show the learned graphs can result in drastically improved efficiency and accuracy in SC tasks. The detailed description of our evaluation metrics, data sets and experiment setup has been provided in Appendix.

Table 1 shows the ACC and NMI results of SC with graphs constructed by different methods with the best numbers highlighted, where graph construction time (Time-C) and spectral clustering time (Time-S) that involves eigendecomposition and kmeans clustering have also been reported. Note that the high computational and memory cost of recent GSP-based graph learning methods, such as GL-SigRep (Dong et al., 2016), GL-Logdet (Dong et al., 2016) and GLSC (Egilmez et al., 2017) do not allow for processing data sets with more than a few thousands of data entities, thus can not be used for real-world SC tasks. We observe that GRASPEL can consistently lead to dramatic performance improvement in SC. Specifically, GRASPEL beats all competitors in clustering accuracy (ACC) across all data sets: GRASPEL achieves more than $18\%$ accuracy gain on USPS and $13\%$ gain on COIL20 over the second-best methods; for the MNIST data set GRASPEL also achieves over $14\%$ accuracy gain over the SC with standard kNN graph and more than $6X$ speedup in graph construction time. Note that the graphs learned by GRASPEL are ultra sparse and have tree-like structures, thereby allowing much faster eigendecompositions in SC when comparing with other methods (Wang & Feng, 2017): the SC of the MNIST data set with standard kNN takes over $6,000$ seconds, which will be dramatically improved to require less than three seconds (over $2,000X$ speedup) using the graph learned by our method (GRASPEL).

The superior performance of GRASPEL is due to the following reasons: **1)** In traditional kNN graphs, all the nodes have the same degrees; as a result, the clustering may strongly favor balanced cut, which may lead to improper cuts in high-density regions of the graph. In contrast, GRASPEL always learns ultra-sparse (tree-like) graphs that only include edges with the largest impact to graph spectral (structural) properties; as a result, the corresponding cuts will always occur in proper regions of the graph, which enables to handle even unbalanced data. **2)** Our approach is less susceptible to noise in similarity calculations, since it only connects two nodes according to its spectral criticality (embedding distortion). Based on matrix perturbation theory (Stewart & Sun, 1990), eigenvectors corresponding to small eigenvalues can be more influenced by noises in similarities, which can lead to accuracy degradation when the number of desired clusters is not small (Hennig et al., 2015). Although the consensus kNN (Premachandran & Kakarala, 2013) attempts to use consensus information for discarding noisy edges, the improvement can still be rather limited since it can be difficult to extract useful consensus information from kNN graphs with small neighborhood sizes.

### 4.2 Results for Graph recovery

We also quantitatively compare graph recovery performance of GRASPEL with state-of-the-art GSP-based graph learning methods, by comparing the graphs learned from observations to the ground

Table 1: Spectral Clustering Results

| Data Set | ACC(%)/ NMI/ Time-C (seconds)/Time-S (seconds) | | | | |
|---|---|---|---|---|---|
| | Standard KNN | ConskNN | LSGL | GRASPEL(Fied.) | GRASPEL(Mult.) |
| COIL-20 | 78.80/ 0.86/ **0.36**/0.37 | 79.86/ 0.86/ 0.54/0.28 | 53.12/0.65/60.22/1.02 | **90.27**/ **0.96**/ 0.40/**0.19** | 85.39/ 0.88/ 0.85 /0.20 |
| PenDigits | 81.12/ 0.80/ **1.25**/0.47 | 84.17/ **0.81**/ 8.59/46.41 | 43.55/0.49/1622/15.64 | **85.96**/ 0.80/ 4.51/**0.27** | 84.12/ 0.80/6.55/0.28 |
| USPS | 68.22/ 0.77/ **2.66**/1.02 | 78.94/ 0.82/ 19.82/74.57 | 32.2/0.31/2598/29.37 | **92.59**/ **0.87**/ 5.19/**0.21** | 90.22/ 0.85/ 8.06/0.21 |
| MNIST | 71.95/ 0.72/ 242.38/6785 | - | - | **81.67**/ **0.75**/ 59.27/**2.90** | 79.05/ 0.74/ 75.38/3.20 |

\- indicates that the method is not capable for handling data sets of this scale.

Table 2: Graph Recovery Results

| Algorithm | The Gaussian graph | | | | The ER graph | | | |
|---|---|---|---|---|---|---|---|---|
| | F-measure | Precision | Recall | NMI | F-measure | Precision | Recall | NMI |
| GL-SigRep | **0.8310** | 0.8120 | 0.8826 | **0.5272** | 0.7243 | 0.6912 | 0.8389 | 0.3600 |
| GL-LogDet | 0.8178 | 0.8193 | 0.8521 | 0.4701 | **0.7378** | 0.6983 | 0.8030 | **0.4012** |
| GLSC | 0.7203 | 0.6901 | 0.9000 | 0.3208 | 0.6609 | 0.5427 | 0.8224 | 0.3379 |
| GRASPEL | **0.8499** | 0.8394 | 0.8812 | **0.5397** | **0.7256** | 0.6990 | 0.8132 | **0.3607** |

truth. The experiments are performed for two widely-used synthetic graphs: 1) The Gaussian graph: the coordinates of the vertices are generated uniformly in the unit square randomly. Edge weights are determined by the Gaussian radial basis function. 2) The ER graph: the graphs generated by following the Erdos-Renyi model (Erdős & Rényi, 1960).

Four widely adopted evaluation metrics in information retrieval have been adopted: Precision, Recall, F-measure and Normalized Mutual Information (Dong et al., 2016). The Precision measures the percentage of correct edges (the edges that are present in the ground-truth graph) in the learned graph. The Recall measures the percentage of the edges in the ground-truth graph that are also in the learned graph. F-measure measures the overall quality by taking both Precision and Recall into account. NMI measures the mutual dependence between the edge sets of the learned graph and the ground-truth graph. The best two F-measure and NMI results have been highlighted in Table 2, showing the effectiveness of GRASPEL in learning graphs that are always close to the ground-truth graphs. Compared with other graph learning methods that can only deal with a few hundreds or thousands of data entities, GRASPEL has much better (nearly-linear runtime and space) scalability and thus will be more efficient for handling large data sets.

As shown in Figure 3, the graph recovery runtime results of GRASPEL has been compared with state-of-the-art graph learning methods, such as the GL-SigRep, GL-LogDet and GLSC algorithm proposed in (Dong et al., 2016; Egilmez et al., 2017). Since the graph learning method LSGL proposed in (Kalofolias & Perraudin, 2019) can not produce comparable quality of graph recovery results, we did not show the runtime results in the figure. As observed, the proposed approach has a much better runtime scalability when comparing with state-of-the-art methods.

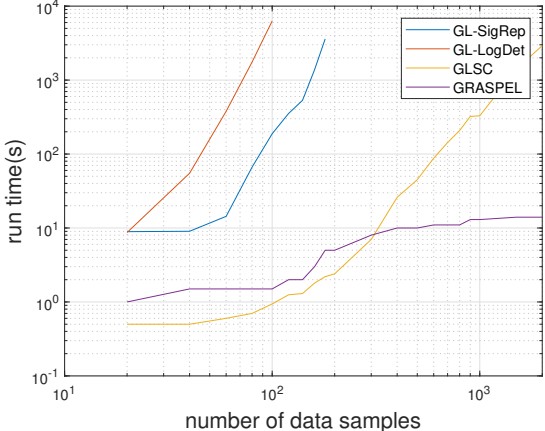

Figure 3: Graph recovery time comparison.

### 4.3 APPLICATIONS IN T-SNE

The t-Distributed Stochastic Neighbor Embedding (t-SNE) has become one of the most popular visualization tools for high-dimensional data analytic tasks (Maaten & Hinton, 2008; Linderman & Steinerberger, 2017). However, its high computational cost limits its applicability to large scale problems. An substantially improved t-SNE algorithm has been introduced based on tree approximation (Van Der Maaten, 2014). However, for large data set the computational cost can still be very high.

A multilevel t-SNE algorithm has been proposed in (Zhao et al., 2018) to dramatically mitigate the computational burden by leveraging spectral graph coarsening as a pre-processing step applied to the original kNN graph. A much smaller set of representative data points can be then selected from the coarsened graph for t-SNE visualization. However, constructing the original kNN graph

can still be costly while the performance may strongly depend on the selection of the number of nearest neighbors. In this work, we use GRASPEL to learn ultra-sparse graphs that can be further reduced into much smaller ones using the spectral graph reduction (Zhao et al., 2018). Then more efficient t-SNE visualization can be achieved based on the data points corresponding to the nodes in the coarsened graphs.

Figure 4 shows the visualization and runtime results of the standard t-SNE (with tree-based acceleration) (Van Der Maaten, 2014) and the multilevel t-SNE algorithm (Zhao et al., 2018) based on graphs learned by GRASPEL. The runtime for the multi-level t-SNE method covers both the graph learning and t-SNE procedures. When using a $5X$ graph reduction ratio, t-SNE can be dramatically accelerated (12.8X and 7X speedups for MNIST and USPS data sets, respectively) without loss of visualization quality.

## USPS data set

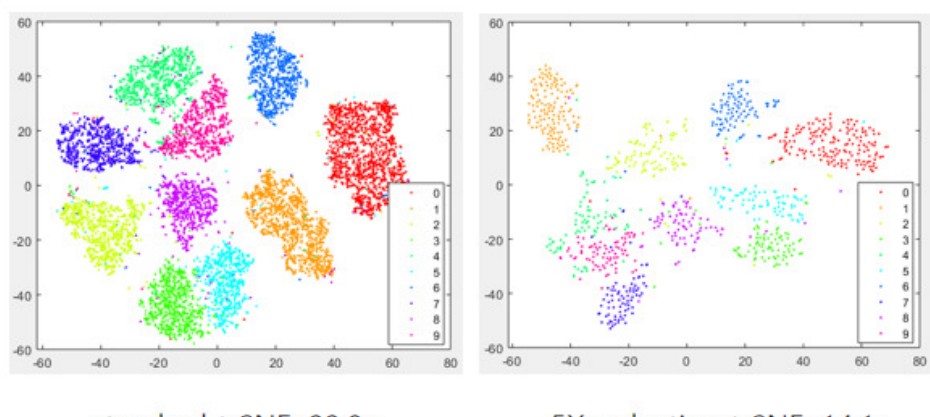

standard t-SNE: 99.8s          5X reduction t-SNE: 14.1s

## MNIST data set

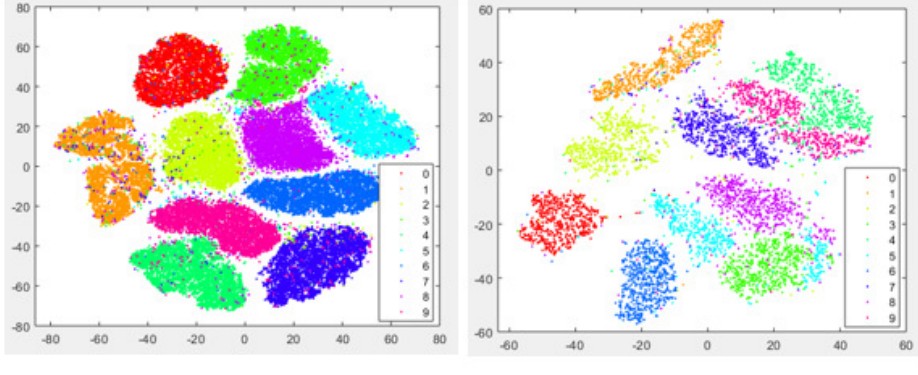

standard t-SNE: 1921.7s          5X reduction t-SNE: 149.3s

Figure 4: t-SNE visualization results.

## 5 CONCLUSION

In this work, we present a scalable spectral approach to graph learning from data. By replacing the precision matrix with a graph Laplacian, our approach aims to estimate ultra-sparse weighted graphs and has a clear connection with the prior graphical Lasso method. Compared with prior graph learning approaches that do not scale to large problems, our approach is more scalable for constructing graphs that can immediately lead to substantially improved computing efficiency and solution quality for a variety of data mining and machine learning applications, such as spectral clustering (SC), and t-Distributed Stochastic Neighbor Embedding (t-SNE).

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

## APPENDIX A    PROOF OF THEOREM 1

Let $L_P$ denote the Laplacian matrix of an undirected graph $P$, and $u_i$ denote the $i$-th eigenvector of $L_P$ corresponding to the $i$-th eigenvalue $\lambda_i$ that satisfies:

$$L_p u_i = \lambda_i u_i, \tag{4}$$

then we have the following eigenvalue perturbation analysis:

$$\left(L_P + \delta L_P\right)\left(u_i + \delta u_i\right) = \left(\lambda_i + \delta \lambda_i\right)\left(u_i + \delta u_i\right), \tag{5}$$

where a perturbation $\delta L_P$ that includes a new edge connection is applied to $L_P$, resulting in perturbed eigenvalues and eigenvectors $\lambda_i + \delta \lambda_i$ and $u_i + \delta u_i$ for $i = 1, ..., n$, respectively.

Keeping only the first-order terms leads to:

$$L_P \delta u_i + \delta L_P u_i = \lambda_i \delta u_i + \delta \lambda_i u_i. \tag{6}$$

Write $\delta u_i$ in terms of the original eigenvectors $u_i$ for for $i = 1, ..., n$:

$$\delta u_i = \sum_{i=1}^{n} \alpha_i u_i. \tag{7}$$

Substituting (7) into (6) leads to:

$$L_P \sum_{i=1}^{n} \alpha_i u_i + \delta L_P u_i = \lambda_i \sum_{i=1}^{n} \alpha_i u_i + \delta \lambda_i u_i. \tag{8}$$

Multiplying $u_i^T$ to both sides of (8) results in:

$$u_i^T L_P \sum_{i=1}^{n} \alpha_i u_i + u_i^T \delta L_P u_i = \lambda_i u_i^T \sum_{i=1}^{n} \alpha_i u_i + \delta \lambda_i u_i^T u_i. \tag{9}$$

Since $u_i$ for for $i = 1, ..., n$ are unit-length, mutually-orthogonal eigenvectors, we have:

$$u_i^T L_P \sum_{i=1}^{n} \alpha_i u_i = \alpha_i u_i^T L_P u_i, \qquad \lambda_i u_i^T \sum_{i=1}^{n} \alpha_i u_i = \alpha_i u_i^T \lambda_i u_i. \tag{10}$$

Substituting (4) into (10), we have:

$$\alpha_i u_i^T L_P u_i = \alpha_i u_i^T \lambda_i u_i. \tag{11}$$

According to (10), we have:

$$u_i^T L_P \sum_{i=1}^{n} \alpha_i u_i = \lambda_i u_i^T \sum_{i=1}^{n} \alpha_i u_i. \tag{12}$$

Substituting (12) into (9) leads to:

$$u_i^T \delta L_P u_i = \delta \lambda_i u_i^T u_i = \delta \lambda_i. \tag{13}$$

Then the eigenvalue perturbation due to $\delta L_P$ is given by:

$$\delta \lambda_i = w_{p,q} \left(u_i^T e_{p,q}\right)^2. \tag{14}$$

If each edge weight $w_{p,q}$ encodes the similarity of data vectors $x_p$ and $x_q$ at nodes $p$ and $q$, it can be shown that $w_{p,q} \propto \frac{1}{z^{data}}$, where $z^{data}$ denotes the distance between $x_p$ and $x_q$; on the other hand, $\left(u_i^T e_{p,q}\right)^2 \propto z^{emb}$. Therefore, as long as we can find an edge with large $w_{p,q} \left(u_i^T e_{p,q}\right)^2$ or $\eta_{p,q} = \frac{z_{p,q}^{emb}}{z_{p,q}^{data}}$, including this edge into the current graph will significantly perturb the Laplacian eigenvalue $\lambda_i$ and eigenvector $u_i$.

## APPENDIX B    ALGORITHMS

---

**Algorithm 1** GRASPEL with Fiedler-vector based spectrally-critical edge identification

---

**Input:** A data set $D$ with $N$ data points $x_1, ... x_N \in \mathbb{R}^d$, window size $\epsilon$, edge selection ratio $\zeta$.
**Output:** The spectrally-learned graph.

1: Construct an initial ANN graph $G_i$ as in (Chen et al., 2011).
2: Initialize: $Terminate$=0;
3: **while** $Terminate$==0 **do**
4:     Embed $G_i$ with the Fiedler vector and sort the data points (nodes);
5:     Evaluate the embedding distortions of candidate edges connecting the top and bottom $\epsilon N$ sorted nodes;
6:     Select top $\zeta N$ edges based on the evaluation result and add them to $G_i$;
7:     Check the spectral stability and update $Terminate$.
8: **end while**

---

**Algorithm 2** GRASPEL with eigenvectors-based spectrally-critical edge identification

---

**Input:** A data set $D$ with $N$ data points $x_1, ... x_N \in \mathbb{R}^d$, $r$ and $l$ that denote the numbers of edges to be added in the spectrally-coarsened graph and the original graph in each iteration, respectively.
**Output:** The spectrally-learned graph.

1: Construct an initial ANN graph $G_i$ as in (Chen et al., 2011);
2: Initialize: $Terminate$=0;
3: **while** $Terminate$==0 **do**
4:     Perform spectral graph coarsening on $G_i$ to obtain its coarsened graph $G_c$;
5:     Spectral clustering of $G_c$ with multiple eigenvectors;
6:     Select $r$ edges with largest distortion between distant clusters of $G_c$;
7:     **for** Each selected edge in $G_c$ **do**
8:         **for** Each of its two nodes **do**
9:             Find its corresponding set of nodes in $G_i$;
10:        **end for**
11:        Form the edge set $E_{orig}$ between the above two sets of nodes;
12:        **if** $|E_{orig}| \leq \lceil \frac{l}{r} \rceil$ **then**
13:            Add all edges $\in E_{orig}$ to $G_i$;
14:        **else**
15:            **for** Each edge $\in E_{orig}$ **do**
16:                Evaluate its embedding distortion;
17:            **end for**
18:            Sort edges $\in E_{orig}$ based on their embedding distortions and add the top $\lceil \frac{l}{r} \rceil$ ones to $G_i$;
19:        **end if**
20:    **end for**
21:    Check the spectral stability and update $Terminate$.
22: **end while**

---

**Algorithm 3** Spectral Clustering Algorithm

---

**Input:** A graph $G = (V, E, w)$ and the number of clusters k.
**Output:** Clusters $C_1 ... C_k$.

1: Compute the adjacency matrix $A_G$, and diagonal matrix $D_G$;
2: Obtain the unnormalized Laplacian matrix $L_G$=$D_G$-$A_G$;
3: Compute the eigenvectors $u_1, ... u_k$ that correspond to the bottom k nonzero eigenvalues of $L_G$;
4: Construct $U \in \mathbb{R}^{n \times k}$, with k eigenvectors of $L_G$ stored as columns;
5: Perform k-means algorithm to partition the rows of $U$ into k clusters and return the result.

---

## APPENDIX C    DATA SETS DESCRIPTION

COIL20: A data set contains $1,440$ gray-scale images of 20 objects, and each object on a turntable has 72 normalized gray-scale images taken from different degrees. The image size is 32x 32 pixels.

PenDigits: A data set consists of 7,494 images of handwritten digits from 44 writers, using the sampled coordination information. Each digit is represented by 16 attributes.

USPS: A data set includes $9,298$ scanned hand-written digits on the envelops from U.S. Postal Service with 256 attributes.

MNIST: A data set consists of 70,000 images of handwritten digits. Each image has 28-by-28 pixels in size. This database can be found from Prof.Yann LeCun's website (http://yann.lecun.com/exdb/mnist/).

## APPENDIX D   COMPARED ALGORITHMS

**Standard kNN**: the most widely used affinity graph construction method. Each node is connected to its $k$ nearest neighbors.

**Consensus of kNN (cons-kNN)** (Premachandran & Kakarala, 2013): adopts the state-of-the-art neighborhood selection methods to construct the affinity graphs. It selects strong neighborhoods to improve the robustness of the graph by using the consensus information from different neighborhoods in a given kNN graph.

**LSGL** (Kalofolias & Perraudin, 2019): a method to automatically select the parameters of the model introduced in (Kalofolias, 2016) given a desired graph sparsity level.

**GL-SigRep** (Dong et al., 2016): construct a graph from signals that are assumed to be smooth with respect to the corresponding graph.

**GL-LogDet** (Dong et al., 2016): encodes the information about the partial correlations between the variables without the constraint to form a valid Laplacian.

**Graph Learning under structural constraints (GLSC)** (Egilmez et al., 2017): formulated the problem as to maximum a posterior estimation of Gaussian Markov Random Field (GMRF) when the precision matrix is chosen to be a graph Laplacian.

## APPENDIX E   EVALUATION METRIC

1)

$$ACC = \frac{\sum\limits_{j=1}^{n} \delta(y_i, map(c_i))}{n}, \tag{15}$$

where $n$ is the number of samples in the data set, $y_i$ is the ground-truth label provided by the data sets, and $c_i$ is clustering result obtained from the algorithm. $\delta(x,y)$ is a delta function defined as: $\delta(x,y)$=1 for $x = y$, and $\delta(x,y)$=0, otherwise. $map(\bullet)$ is a permutation function that maps each cluster index $c_i$ to a ground truth label, which can be realized using the Hungarian algorithm (Papadimitrou & Steiglitz, 1982). ACC measures the agreement between the clustering results generated by clustering algorithms and the ground-truth labels. A higher value of $ACC$ indicates better clustering quality.

2)

For two random variables $P$ and $Q$, normalized mutual information is defined as (Strehl & Ghosh, 2002):

$$NMI = \frac{I(P,Q)}{\sqrt{H(P)H(Q)}}, \tag{16}$$

where $I(P,Q)$ denotes the mutual information between $P$ and $Q$, while $H(P)$ and $H(Q)$ are entropies of $P$ and $Q$. In practice, the NMI metric can be calculated as follows (Strehl & Ghosh, 2002):

$$NMI = \frac{\sum\limits_{i=1}^{k}\sum\limits_{j=1}^{k} n_{i,j} \log(\frac{n \cdot n_{i,j}}{n_i \cdot n_j})}{\sqrt{(\sum\limits_{i=1}^{k} n_i \log \frac{n_i}{n})(\sum\limits_{j=1}^{k} n_j \log \frac{n_j}{n})}}, \tag{17}$$

where $n$ is the number of data points in the data set, k is the number of clusters, $n_i$ is the number of data points in cluster $C_i$ according to the clustering result generated by algorithm, $n_j$ is the number of data points in class $C_j$ according to the ground truth labels provided by the data set, and $n_{i,j}$ is the number of data points in cluster $C_i$ according to the clustering result as well as in class $C_j$ according to the ground truth labels. The NMI value is in the range of [0, 1], while a higher NMI value indicates a better matching between the algorithm generated result and ground truth result.

