# OpenReview forum: "GRASPEL: GRAPH SPECTRAL LEARNING AT SCALE"
_ICLR.cc/2020/Conference — Reject_

### Official Review · AnonReviewer3 · 2019-10-23
**Official Blind Review #3**

**Rating:** 3

**Review:**

This paper proposes a scalable approach for graph learning from data. At a high-level, it begins with a k-NN graph construction, then node features are embedded to spectral space (embedding to space spanned by eigenvectors of Laplacian). Next, edges that have a large distortion are additionally added to the latest graph. And these steps are repeated until the output graph is stable (i.e., the embedding distortion becomes small). Experimental result for spectral clustering shows that the proposed method can achieve the best accuracy compared to kNN-based methods. For graph recovery, the algorithm also performs better than other Laplacian-based graph learning methods. In addition, the proposed approach runs up to 5 times faster for t-SNE.

The authors demonstrate that their algorithm is scalable and faster than Laplacian-based methods requiring O(N^2). However, the proposed method also requires to compute eigenvectors of Laplacian thus it seems not to be faster compared to the previous algorithm. It would be better to provide the time complexity of each step in section 3.2 and that of the overall algorithm.

It is unclear that the proposed algorithm (section 3.2) is optimized the objective function in equation (9). And it is possible to theoretically guarantee that the algorithm finds a spectrally optimized graph?

For experiments, although the authors argue that the proposed algorithm is scalable, datasets that they used are not large-scale. And it is needed to provide runtimes of other algorithms for graph recovery tasks (section 4.2).

Overall, this paper develops a new approach, but its novelty and intuition are unclear. Moreover, it does not seem to be scalable under the bar of acceptance.

Minor concerns:
There is no content in section 3.2.5.


**Experience Assessment:**

I have read many papers in this area.

**Review Assessment: Checking Correctness Of Derivations And Theory:**

I assessed the sensibility of the derivations and theory.

**Review Assessment: Checking Correctness Of Experiments:**

I assessed the sensibility of the experiments.

**Review Assessment: Thoroughness In Paper Reading:**

I read the paper at least twice and used my best judgement in assessing the paper.

---

> ### Author Response · Authors · 2019-11-15
> **Response to Reviewer 3**
>
> Q1. The authors demonstrate that their algorithm is scalable and faster than Laplacian-based methods requiring O(N^2). However, the proposed method also requires to compute eigenvectors of Laplacian thus it seems not to be faster compared to the previous algorithm. It would be better to provide the time complexity of each step in section 3.2 and that of the overall algorithm.
>
> A1: Thanks for pointing this out. Our spectral graph embedding leverages a recent spectral graph coarsening approach to achieve nearly-linear time complexity for computing the first few graph Laplacian eigenvalues and eigenvectors.  We note that this is the first work that introduces a spectral method for scalable graph learning from data by leveraging the latest results in spectral graph theory. Since in the proposed work all the kernel functions, such as spectral graph sparsification, spectral graph coarsening and spectral graph embedding methods are nearly-linear time algorithms, the entire spectral graph learning approach is also highly scalable. We have included more results to show the scalability of our method and stressed the above fact in the revised draft.
>
>
> Q2: It is unclear that the proposed algorithm (section 3.2) is optimized for the objective function in equation (9). And it is possible to theoretically guarantee that the algorithm finds a spectrally optimized graph?
>
> A2: This is a very good suggestion. In the revised paper, we have included a description of the connection between our algorithm and the optimization objective in (2). The original optimization objective function (9) includes three components: (a) log (det L) that corresponds to the sum of the Laplacian eigenvalues, (b) - \alpha* X^T L X that corresponds to the smoothness of signals across the graph, and (c) - \beta* |L|_0 that corresponds to graph sparsity. Our algorithm flow aims to iteratively identify and include the most spectrally-critical edges into the latest graph so that the first few Laplacian eigenvalues & eigenvectors can be most significantly perturbed with the minimum amount of edges. Since the inclusion of spectrally-critical edges will immediately improve distortion in the embedding space, the overall smoothness of graph signals will thus be significantly improved. In other words, the spectrally-critical edges will only impact the first few Laplacian eigenvalues and eigenvectors key to graph spectral properties, but not the largest few eigenvalues and eigenvectors-which will require adding much more edges to influence. It can be easily shown that including any additional edge into the graph will monotonically increase (a), but monotonically decrease (b) and (c). Specifically, when the spectra of the learned graph is not stable, adding spectrally-critical edges will dramatically increase  (a), while decreasing (b) and (c) at a much lower rate since the improved graph signal smoothness will only result in a slight change (increase) to  Tr(X^T L x). Consequently, the objective function in (2) will be effectively maximized by including only a small amount of spectrally-critical edges until the first few eigenvalues become sufficiently stable; when adding extra edges can no longer significantly perturb the first few eigenvalues, (b) and (c) will start to dominate the objective function value, indicating that the iterations should be terminated. The stopping condition can be controlled by properly setting an embedding distortion threshold for $\eta $ or parameters $\alpha$  and $\beta$. We have included the above convergence analysis in the revised draft.
>
> Q3: For experiments, although the authors argue that the proposed algorithm is scalable, datasets that they used are not large-scale. And it is needed to provide runtimes of other algorithms for graph recovery tasks (section 4.2).
>
> A3: Thanks for the suggestion. We have compared our methods with state-of-the-art graph learning methods published in ICLR’19 paper "Large scale graph learning from smooth signals." by Kalofolias, Vassilis, and Nathanaël Perraudin. As shown, our approach is over 400X faster for graph construction while achieving consistently much better accuracy in spectral clustering tasks. We also added a figure (Figure 3) showing the scalabilities of comparisons with state-of-the-art methods for graph recovery tasks.

---

### Official Review · AnonReviewer1 · 2019-10-24
**Official Blind Review #1**

**Rating:** 6

**Review:**

This paper presents a scalable spectral approach for graph learning. In particular, the authors use graph Laplacian as precision matrix, and show the connection between the proposed method and graphical Lasso. Three tasks, including spectral clustering, graph recovery and t-SNE visualization, are considered in experiments.

Pros.
1. Scalable graph learning is an important research topic. This paper presents a practical solution to large-scale graph learning.
2. The connection between the proposed method and graphical Lasso is discussed. Also, theoretical analysis on spectral criticality is provided.
3. Overall the paper is well organized and clearly written.

Cons.
1. My major concern is about the experiments. The authors claim that the proposed graph learning approach is highly scalable. It would be more convincing if the authors can evaluate the proposed method on larger datasets.
2. One of the tasks in experiments is t-SNE visualization. There are also some faster versions of t-SNE with a complexity of O(NlogN), such as [a]. For t-SNE, the authors may justify what's the advantage of using the proposed method over other fast t-SNE algorithms.
[a] Accelerating t-SNE using Tree-Based Algorithms, JMLR 2014.

**Experience Assessment:**

I have read many papers in this area.

**Review Assessment: Checking Correctness Of Derivations And Theory:**

I assessed the sensibility of the derivations and theory.

**Review Assessment: Checking Correctness Of Experiments:**

I carefully checked the experiments.

**Review Assessment: Thoroughness In Paper Reading:**

I read the paper at least twice and used my best judgement in assessing the paper.

---

> ### Author Response · Authors · 2019-11-15
> **Response to Reviewer 1**
>
> Q1. My major concern is about the experiments. The authors claim that the proposed graph learning approach is highly scalable. It would be more convincing if the authors can evaluate the proposed method on larger datasets.
>
> A1: Thanks for the suggestion. We have compared our methods with state-of-the-art graph learning methods published in ICLR’19 paper "Large scale graph learning from smooth signals." by Kalofolias, Vassilis, and Nathanaël Perraudin. As shown, our approach is over 400X faster for graph learning while achieving consistently much better accuracy in spectral clustering tasks. We also added Figure 3 to show the runtime comparisons with state-of-the-art methods for graph recovery tasks.
>
> Q2. One of the tasks in experiments is t-SNE visualization. There are also some faster versions of t-SNE with a complexity of O(NlogN), such as [a]. For t-SNE, the authors may justify what's the advantage of using the proposed method over other fast t-SNE algorithms.
> [a] Accelerating t-SNE using Tree-Based Algorithms, JMLR 2014.
>
> A2: Thanks very much for the suggestion. Our results (standard t-SNE) reported in the paper are obtained by using the tree-based t-SNE algorithm that is a default option in Matlab. We have clarified this in the revised paper.

---

### Official Review · AnonReviewer2 · 2019-10-27
**Official Blind Review #2**

**Rating:** 1

**Review:**

In this paper, the authors present a method that transforms data into graph. They emphasize on the fact that the proposed method is scalable, using a spectral embedding to construct the graph.

We think that the paper is not of enough quality to be accepted in ICLR. Without going in detail in the derivations, we give below some major issues in this submitted paper.

The studied problem has been widely investigated in the literature. Many methods have been proposed within the same objective, including taking care of the scalability issue. The authors fail to provide the state of the art, as well as describe the contributions with respect to previous work. As a consequence, the contributions are not clear. Maybe the proposed framework is original, but the there has been plenty of methods that have considered the same problem.

Experiments are poor and not convincing. The authors compare the proposed method to only two spectral clustering methods, which as the standard kNN and the Consensus kNN from 2013. These two methods are pretty old and many more recent methods have been introduced in the literature. Moreover, the results in Table 1 are somehow misleading, as the standard kNN is faster that the proposed method on 3 out of 4 datasets. Experiments in graph recovery are not clear, starting from the fact that the datasets are not defined (what are the Gaussian graph and ER graph?), neither the experimental setting (what is the problem at hand?). The same goes to the application of t-SNE which is also very weak.

--------------
Reply to Rebuttal

The authors have modified the paper to take into consideration our previous comments and suggestions. However, we think that it is still of not sufficient quality. We give below some elements, without providing a thorough review.

It is pretty pretentious to say that "this is the first work that introduces a spectral method for learning ultra-sparse (tree-like) graphs from data", while not comparing to the state of the art. There have been many spectral methods in graph learning for large-scale datasets.

In experiments, the only added method is the one of Kalofolias and Perraudin (submitted in 2017 to ArXiv). However, results show that this method is the worst of all methods. It is even the worst compared to the simple standard knn. It is not clear how the authors get such results; It looks like something is wrong in experiments, or they are cherrypicking.


**Experience Assessment:**

I have published in this field for several years.

**Review Assessment: Checking Correctness Of Derivations And Theory:**

I assessed the sensibility of the derivations and theory.

**Review Assessment: Checking Correctness Of Experiments:**

I carefully checked the experiments.

**Review Assessment: Thoroughness In Paper Reading:**

I read the paper at least twice and used my best judgement in assessing the paper.

---

> ### Author Response · Authors · 2019-11-15
> **Response to Reviewer 2**
>
> Q1. The studied problem has been widely investigated in the literature. Many methods have been proposed within the same objective, including taking care of the scalability issue. The authors fail to provide the state of the art, as well as describe the contributions with respect to previous work. As a consequence, the contributions are not clear. Maybe the proposed framework is original, but there has been plenty of methods that have considered the same problem.
>
> A1: Thanks very much for pointing this out. We have added descriptions regarding our contribution to the abstract and introduction. Our contribution: this is the first work that introduces a spectral method for learning ultra-sparse (tree-like) graphs from data by leveraging the latest results in spectral graph theory, such as the nearly-linear-time spectral graph sparsification, spectral coarsening and spectral embedding techniques. Our framework is similar to the original graphical Lasso framework with the precision matrix replaced by a graph Laplacian matrix. This approach iteratively identifies and includes the most spectrally-critical edges into the latest graph, so that the first few Laplacian eigenvalues and eigenvectors can be most significantly perturbed by including the minimum amount of edges. The iterations will be terminated when the graph spectra become sufficiently stable (or graph signals become sufficiently smooth across the graph and lead to rather small Laplacian quadratic forms). High-quality estimation of attractive Gaussian Markov Random Fields (GMRFs) can be achieved for much larger datasets compared with state-of-the-art methods. The graphs learned from our approach allow obtaining much more accurate results more efficiently in spectral clustering tasks (due to the ultra-sparse tree-like structure) and faster performance for t-SNE visualization of large data sets. We have also included more details about the connection between our algorithm with the original optimization objective in (2) in the revised draft.
>
> Q2. Experiments are poor and not convincing. The authors compare the proposed method to only two spectral clustering methods, which as the standard kNN and the Consensus kNN from 2013. These two methods are pretty old and many more recent methods have been introduced in the literature. Moreover, the results in Table 1 are somehow misleading, as the standard kNN is faster that the proposed method on 3 out of 4 datasets. Experiments in graph recovery are not clear, starting from the fact that the datasets are not defined (what are the Gaussian graph and ER graph?), neither the experimental setting (what is the problem at hand?). The same goes to the application of t-SNE which is also very weak.
>
> A2: Thanks very much for the kind suggestion. GRASPEL indeed runs slightly slower than the standard kNN for very small datasets but much faster for larger ones. More importantly, the graphs learned by our approach have ultra-sparse tree-like structures (the edge to node ratio is between 1.1 to 1.3) and will result in significantly improved accuracy and efficiency in spectral clustering. As shown in Table 1 that includes substantially updated results, the spectral clustering time for the MNIST data set with standard kNN is over 6000 seconds but will be dramatically brought down to less than three seconds (over 2000X speedup) using the graph learned by our method (GRASPEL). We also have compared our methods with state-of-the-art graph learning methods published in ICLR’19, "Large scale graph learning from smooth signals." by Kalofolias, Vassilis, and Nathanaël Perraudin. As shown, our approach is over 400X faster for graph construction while achieving consistently much better accuracy in spectral clustering tasks.  We also added Figure 3 to show the runtime comparisons with state-of-the-art methods for graph recovery tasks.

---

> ### Public Comment · ~Stacy_X_Pu1 · 2020-01-31
> **To Reviewer2: What are the state-of-the-art methods? Could you cite the paper here?**
>
> Hi Reviewer2,
>
> I would like to ask what research papers, in your opinion, addressing the same problem (learning graphs at scale)? Could you name some papers with the state-of-the-art methods here?
>
> Many Thanks

---

### Author Response · Authors · 2019-11-15
**Summary of our update**

Thanks for the comments from all three reviewers. We added additional experiments and clarification into our modified paper (marked in blue). Specifically, (1) we completely rewrote Sections 2 and 3, as well as modified Sections 1 and 4 to more clearly highlight our contribution and results; (2) convergence and complexity analysis has also been included into Section 3; (3) we added additional experimental results comparing with the ICLR’19 paper ("Large scale graph learning from smooth signals.") and included additional runtime results for spectral clustering tasks using the graphs learned (constructed) by different methods; (4) we also demonstrated GRASPEL’s runtime scalability for graph recovery tasks in Figure 3 by comparing it with state-of-the-art methods on data sets of different sizes.

---

### Decision · Program_Chairs · 2019-12-19

**Decision:**

Reject

**Comment:**

This paper proposes a scalable approach for graph learning from data. The reviewers think the approach appears heuristic and it is not clear the algorithm is optimizing the proposed sparse graph recovery objective.